# Transmission of H9N2 Low Pathogenicity Avian Influenza Virus (LPAIV) in a Challenge-Transmission Model

**DOI:** 10.3390/vaccines10071040

**Published:** 2022-06-28

**Authors:** Sugandha Raj, Jake Astill, Nadiyah Alqazlan, Nitish Boodhoo, Douglas C. Hodgins, Éva Nagy, Samira Mubareka, Khalil Karimi, Shayan Sharif

**Affiliations:** 1Department of Pathobiology, Ontario Veterinary College, University of Guelph, Guelph, ON N1G 2W1, Canada; rajs@uoguelph.ca (S.R.); nadiyah.alqazlan@gmail.com (N.A.); boodhoon@uoguelph.ca (N.B.); dhodgins@uoguelph.ca (D.C.H.); enagy@ovc.uoguelph.ca (É.N.); kkarimi@uoguelph.ca (K.K.); 2Artemis Technologies Inc., Guelph, ON N1L 1E3, Canada; jake.s.astill@gmail.com; 3Divisions of Infectious Diseases and Microbiology, Sunnybrook Research Institute, Toronto, ON M4N 3M5, Canada; samira.mubareka@sunnybrook.ca

**Keywords:** chickens, influenza, virus, H9N2, transmission, infection, shedding

## Abstract

Migratory birds are major reservoirs for avian influenza viruses (AIV), which can be transmitted to poultry and mammals. The H9N2 subtype of AIV has become prevalent in poultry over the last two decades. Despite that, there is a scarcity of detailed information on how this virus can be transmitted. The current study aimed to establish a direct contact model using seeder chickens infected with H9N2 AIV as a source of the virus for transmission to recipient chickens. Seeder chickens were inoculated with two different inoculation routes either directly or via the aerosol route. The results indicate that inoculation via the aerosol route was more effective at establishing infection compared to the direct inoculation route. Shedding was observed to be higher in aerosol-inoculated seeder chickens, with a greater percentage of chickens being infected at each time point. In terms of transmission, the recipient chickens exposed to the aerosol-inoculated seeder chickens had higher oral and cloacal virus shedding compared to the recipient chickens of the directly inoculated group. Furthermore, the aerosol route of infection resulted in enhanced antibody responses in both seeder and recipient chickens compared to the directly inoculated group. Overall, the results confirmed that the aerosol route is a preferred inoculation route for infecting seeder chickens in a direct contact transmission model.

## 1. Introduction

Avian influenza viruses (AIV) are enveloped, negative sense, single-stranded RNA viruses with a segmented genome belonging to the family *Orthomyxoviridae* [1]. The H9N2 AIV subtype is a re-emerging virus that is endemic in countries in the Middle East, Central Asia, and Africa, with sporadic outbreaks reported from Hong Kong and many provinces of China over the last two decades [2,3]. AIV is categorized into high and low pathogenicity avian influenza viruses (HPAIV and LPAIV, respectively) [4]. The H9 subtypes of influenza viruses are LPAIV. Infection with H9N2 AIV is reported to cause mildly clinical to subclinical infections with a low mortality rate, but can cause economic losses in the poultry industry associated with marked reduction in meat and egg production and decreased body weight [2]. This virus induces mild immunosuppression which can aggravate secondary bacterial and viral co-infections in poultry, often causing fatalities [5,6].

The H9N2 AIV subtype possesses the potential to spread across large geographic distances. A high mutation rate and the geographic spreading of avian influenza viruses have led to the evolution of five phylogenetically characterized subgroups, identified by specific fingerprint tools [7]. The migratory wild waterfowl act as major reservoirs and can transmit LPAIVs to avian and mammalian species, such as pigs, whales, seals, and horses [3,8]. Although transmission of H9N2 AIV from avian species to humans has not been widely reported, it was isolated in humans in 2003, 2016, and 2017 in Hong Kong, China [9,10]. Additionally, some recent studies have shown the aerosol transmission of H9N2 AIV mutants can occur from avian to mammalian species [11]. A total of 58 human infections of H9N2 AIV have been reported to WHO between 2015–2021 [12]. The adaptability of H9N2 AIVs in humans can be associated with its ability to undergo constant mutation(s) in its genomic segments, to overcome selective pressure, and subsequently cause outbreaks in humans, causing a public health risk. The ability of the virus to donate its genomic segments to other subtypes has been linked to outbreaks of H7N7, H7N9 (2012) and, more recently, H5N6 in China [13]. This demonstrates the zoonotic potential of H9 AIV to infect humans and highlights the need for rapid and sensitive detection methods (molecular fingerprinting and biological adaptation markers) for AIV surveillance in poultry and mammals [14,15]. 

One of the major concerns associated with H9N2 AIV infections in poultry is the high transmission rate of the virus, leading to dissemination of the virus in the environment. Transmission of AIVs is thought to occur via inhalation of droplet nuclei and aerosols, direct or indirect exposure to infected birds, contact with fomites (contaminated objects in the environment), and the oral-fecal route [16,17,18,19]. 

A previous study by Guan and colleagues (2013) revealed that aerosol transmission of H9N2 AIV was more efficient than the direct or indirect routes of transmission in chickens. Aerosol transmission led to higher infection in recipient (naïve) chickens with a lower inoculation dose than that by intranasal inoculation in direct or indirect contact models [16]. Therefore, the present study aimed to determine the relative impact of two different routes of inoculation on transmission of H9N2 AIV in chickens focused in a direct contact transmission model. The study was performed by cohousing recipient chickens with seeder chickens infected via the aerosol route or by direct inoculation (a combination of ocular, nasal, and intratracheal routes). Additionally, we sought to investigate the route of inoculation that provides the highest amount of virus shedding in the inoculated seeder chickens. Finally, the ability of viruses shed from infected seeder chickens to cause infection and induce an immune response in the recipient chickens was determined.

## 2. Materials and Methods

### 2.1. Chickens

A total of seventy-two one-day-old, specific-pathogen-free (SPF), white Leghorn chickens were obtained from the Canadian Food Inspection Agency (Ottawa, ON, Canada). Birds were grouped in the Horsfall units (4 ft × 4 ft × 4 ft) in the Research Isolation Unit at the University of Guelph. All of the experiments and procedures were approved by the Animal Care Committee at the University of Guelph and complied with specifications of the Canadian Council on Animal Care.

### 2.2. Virus Propagation and Infectious Dose 

An H9N2 LPAIV strain, A/TK/IT/13VIR1864-45/2013 exhibiting a Korean lineage was selected for the present research. The virus strain was generously provided by the Instituto Zooprofilattico Spermentale delle Venezie (IZSVe), Italy. For virus propagation, 10-day-old embryonated chicken eggs were inoculated with the virus and incubated for a period of 72 h at 37 °C. Seventy-two hours post-incubation, the eggs were maintained overnight at 4 °C. The allantoic fluid was then collected and centrifuged at 400× *g* for 15 min and stored at −80 °C. The virus titers were quantified by titrating the virus on Madin-Darby canine kidney (MDCK) cells. The titers were based on the end-point dilutions expressed as 50% tissue culture infectious dose (TCID_50_/mL) [20]. For the present study, virus inoculum containing 8 × 10^8^ TCID_50_ of H9N2 AIV in 250 μL was used to infect the inoculated seeder chickens in both the experimental setups. A defined virus dose for a sustainable transmission of LPAIVs remains unknown, due to variations in adaptability of LPAIVs to different species [21]. Previous studies have revealed that the replication rate of AIV differs among species, usually being higher in the species of origin [21,22,23]. Since the virus used in this study had a turkey origin, we conducted several pilot experiments to establish transmission and decided to use a higher inoculation dose to ensure infection and transmission to healthy recipient chickens. 

### 2.3. Experimental Design 

A direct contact transmission model for H9N2 AIV was established using two different routes of infection (direct inoculation and aerosol). For each route of inoculation, the experimental setup was comprised of two subgroups: an inoculated seeder group (n = 16) and a recipient group (n = 8) (healthy and uninfected). For proper identification, the chickens were tagged using poultry wing bands (tags) to differentiate between the inoculated seeder and recipient groups. On day 14 of age, the seeder chickens in each setup were inoculated with H9N2 AIV either through direct inoculation or the aerosol route, or with phosphate buffer saline (PBS) in the control group.

For infection via direct inoculation route, the seeder chickens were infected via a combination of ocular, nasal, and intratracheal routes (50 μL/each route; direct inoculation group). To evaluate the infection via the aerosol route (aerosol group), the virus was aerosolized in an aerosol chamber. The aerosol chamber had a volume of 20,000 cm^3^ (40 cm × 20 cm × 25 cm) and was maintained inside a biological hood for the experiment. Three seeder chickens at a time were placed inside the aerosol chamber and 750 μL of H9N2 AIV inoculum (250 μL/chicken) was added to the nebulizer (PARI LC^®^ Sprint Nebulizer Set, PARI Respiratory Equipment, Inc., Midlothian, UK) and aerosolized by an air compressor (ProNebultra, PriRespiratory Equipment Inc., Monterey, CA, USA). The aerosol chamber was coupled with the nebulizer with a 22 mm diameter tube to transport aerosol particles into the chamber. The particle size or the mass median diameter (MMD) of the aerosols generated ranged between 3–3.5 μm, with a total output rate (TOR) of 500 mg/min, according to the manufacturer’s information. The seeder chickens were held in the aerosol chamber for a period of 20 min for maximum inhalation of aerosol particles [24]. Twenty-four hours post-infection, the recipient (healthy and uninfected) chickens were grouped with the respective inoculated seeder chickens in each model. Both the inoculated seeder and recipient groups were housed together for a period of 14 days after grouping of the recipient chickens with the inoculated seeder chickens (Appendix A). The total duration of the experiment was 29 days.

It should be noted that the actual dose inhaled by the aerosol-inoculated seeder chickens was difficult to quantify in the present study. This is due to a difference in the rate of inhalation by each chicken, which affects the overall uptake of H9N2 AIV over time. Additionally, the deposition of aerosolized virus particles in different areas, such as the chicken’s body (feathers, shanks, etc.) or inanimate objects in the cage (water fonts, bedding, feeders), might also contribute to the indirect transmission of the H9N2 AIV [25]. Therefore, the model used in the present study monitored an overall H9N2 AIV transmission, which occurred via aerosol, large droplet, and fomite-based routes of transmission. 

### 2.4. Virus Isolation and Processing of Swab Samples

Chickens in all treatment groups were swabbed for oral and cloacal shedding on days 3, 5, 7, and 9 post-infection (PI) and post-exposure (PE), respectively. Swab samples were collected in 1.5 mL; centrifuge tubes containing transport medium DMEM (Dulbecco’s Modified Eagle’s medium) were supplemented with 0.5% BSA fraction V, 10 mL Penicillin (200 U/mL) with streptomycin (80 μg/mL), and 5ml gentamycin (50 μg/mL) to prevent any bacterial contamination. The swab samples were maintained throughout at 4 °C on ice from the initial steps of sample collection until processing and virus isolation. Processing of the swab samples involved thorough vortexing for one minute followed by centrifugation at 550× *g* for 10 min at 4 °C. Supernatant was aliquoted and stored at −80 °C until its later use in virus isolation [26]. The TCID50 assay was done according to instructions in the WHO (2011) manual, with slight modifications [26]. The swab samples were titrated by adding them over a confluent (75–90%) monolayer of MDCK cells. MDCK cells were suspended in fresh complete DMEM medium containing 2 μg/mL of L-1-tosylamido-2-phenylethyl chloromethyl ketone (TPCK)-treated trypsin (Sigma-Aldrich, Oakville, ON, Canada; Cat.bNo. T1426). Virus titer in the swabs was quantified using an end-point dilution on MDCK cells and expressed as TCID50/mL using the formula by Reed–Muench [20]. The minimal limit of detection for the assay was 1.78 × 10^2^ TCID50/mL.

### 2.5. Hemagglutination Inhibition (HI) Assay

Serum samples were used to determine the mean antibody levels on day, 7 and 14 PI/PE through the HI assay. A two-fold serial dilution using 50 μL serum samples was performed. An amount of 50 μL of H9N2 AIV preparation containing 8 hemagglutinin units were added, followed by 30 min of incubation at room temperature (RT) in 96-well V bottom plates (Corning Inc., Corning, New York, NY, USA). Chicken red blood cells (RBCs) were then added at 0.5% and the plates were further incubated for 30 min at RT. The HI titer was determined as the reciprocal of the greatest dilution that showed complete inhibition of red blood cell agglutination (log_2_ scale). Rates of seroconversion and seroprotection were defined as post-vaccination HI titers that increased by four-fold or greater, and HI titers > 40, respectively [27]. Seroconversion and seroprotection rates were calculated for all groups on days 7 and 14 PI/PE.

### 2.6. Statistical Analysis

Statistical analyses were performed using the Mann–Whitney nonparametric unpaired Student’s *t*-test using GraphPad Prism 9.0 software. Data was considered statistically significant between treatment groups when *p* < 0.05. Seroconversion rates and seroprotection rates were calculated based on HI antibody titers and were compared using Fisher’s exact test. Correlation between HI titers (log_2_ scale) and virus load (expressed in TCID50/mL) was determined using the Spearman correlation test in the recipient chickens that were positive for H9N2 AIV at different time points of swab collection (days 3, 5, and 7 PE). A two-sided alpha level of 0.05 was considered significant. 

## 3. Results

All chickens were swabbed post-inoculation to determine the viral shedding at different sampling points (days 3, 5, 7, and 9 PI/PE). Serum samples were collected on days 7 and 14 PI/PE to determine the antibody-mediated responses. All experimental birds remained asymptomatic and did not show any clinical signs during the entire experimental period. The control (PBS-inoculated and exposed recipient) chickens remained negative for the virus during the entire course of the experiment. 

### 3.1. H9N2 AIV Infection within the Inoculated Seeder Groups

The results from the present study demonstrated that chickens could be infected by H9N2 AIV via the aerosol and the direct inoculation route. The aerosol-inoculated seeder chickens demonstrated higher virus shedding in both the oral and cloacal swabs on days 3, 5, 7, and 9 PI, with a longer virus shedding period up to day 9 PI, compared to the directly inoculated seeder chickens, who shed the virus until day 7 PI (Figure 1).

The peak in oral shedding was observed on day 3 PI for both the aerosol-inoculated seeder (1.2 × 10^5^ TCID_50_/mL) and directly inoculated seeder groups (7.1 × 10^4^ TCID_50_/mL). The number of chickens orally shedding H9N2 AIV was higher in the aerosol-inoculated seeder group (15/16) when compared to the directly inoculated seeder group (14/16) on day 3 PI. A similar trend was also observed on days 5 (13/16) and 7 PI (10/16), with the aerosol-inoculated seeder group showing a greater number of chickens positive for AIV infection when compared to the directly inoculated seeder group (Table 1). The results of oral virus load indicated a decline in virus shedding from day 5 PI onwards, yet a relatively higher virus load was observed on days 5, 7, and 9 PI (*p* < 0.05) in the aerosol-inoculated seeder group compared to the directly inoculated seeder group (Figure 1A). 

In all treatment groups, the cloacal virus load was higher compared to the oral virus load irrespective of the sampling time points. The results of the cloacal virus load demonstrates a trend with higher virus titers in the aerosol-inoculated seeder chickens on days 3, 5, 7, and 9 PI, with a significant difference in shedding on days 3 and 9 PI when compared to the directly inoculated seeder group (*p* < 0.05) (Figure 1B). The maximum amount of virus load in cloacal samples was detected on day 3 PI, with an average virus load of 4.4 × 10^5^ TCID_50_/mL in the aerosol-inoculated seeder group and 2.4 × 10^5^ TCID_50_/mL in the directly inoculated seeder group. 

The number of chickens positive for virus isolation in the cloacal swabs was higher in the aerosol-inoculated seeder group, with 15/16 chickens positive on day 3 PI compared to 11/16 chickens in the directly inoculated seeder group. The number of virus positive chickens declined from day 3 to day 9 PI in the aerosol-inoculated seeder group; 12/16, 10/16, and 8/16 chickens were positive for virus isolation on days 5, 7, and 9 PI, respectively, compared to 10/16, 5/16, and 0/16 chickens positive on days 5, 7, and 9 PI in the directly inoculated seeder group, respectively (Table 1). 

### 3.2. Transmission of H9N2 AIV in Recipient Chickens

The results presented here demonstrate that the horizontal transmission of H9N2 AIV in the recipient chickens when grouped together with the inoculated seeder chickens could occur via contact-dependent transmission. There was higher virus load in oral and cloacal swabs of the recipient chickens exposed to the aerosol-inoculated seeder group at different time points compared to the recipient chickens exposed to the directly inoculated seeder group. The peak in virus shedding was observed on day 3 PE and the rate of virus shedding declined from day 5 to day 7 PE in both the recipient groups (Figure 2). No virus shedding was detected after day 7 PE in either of the recipient groups. 

With respect to oral virus load, the recipient chickens exposed to the aerosol-inoculated seeder chickens had higher loads of H9N2 AIV at all time points compared to the recipient chickens exposed to the directly inoculated seeder chickens (Figure 2A). Virus load in the oral swabs was detectable as early as day 3 PE in both the recipient groups (aerosol and direct inoculation). Oral virus load in the recipient chickens exposed to the aerosol-inoculated seeder group was determined to be significantly higher (*p* < 0.05) on day 3 PE with an average virus titer of 4.8 × 10^4^ TCID_50_/mL compared to 3.4 × 10^4^ TCID_50_/mL in the recipient chickens exposed to the directly inoculated seeder group (Figure 2A). A similar trend in virus load was observed on day 5 PE in which the recipient chickens of the aerosol group had a significantly higher (*p* < 0.05) average virus load of 2.6 × 10^4^ TCID_50_/mL compared to 9.4 × 10^3^ TCID_50_/mL in the recipient chickens of the directly inoculated seeder group. Although the virus load detected on day 7 PE (9.1 × 10^3^ TCID_50_/mL) was five-fold higher in the recipient chickens exposed to the aerosol-inoculated seeder group, it did not differ significantly from that of the recipient chickens exposed to the directly inoculated seeder group (Figure 2A). 

Considering cloacal shedding, the recipient chickens exposed to the aerosol-inoculated seeder group had a higher number of chickens positive for virus isolation on days 3 (7/8), 5 (7/8), and 7 (6/8) PE (Table 2), with significantly higher virus loads on day 3 (1.3 × 10^5^ TCID_50_/mL) and day 5 PE (9.2 × 10^4^ TCID_50_/mL) compared to recipient chickens exposed to the directly inoculated seeder group (*p* < 0.05) (Figure 2B).

### 3.3. HI Antibody Titers, Seroconversion and Seroprotection

The chickens that were infected either by aerosol or direct inoculation routes (including both inoculated seeder and recipient groups) had detectable antibodies on days 7 and 14 PI/PE. The results also indicated a difference in HI titers on days 7 and 14 PI between the inoculated seeder and recipient groups. An enhanced antibody response was observed on day 14 PI for both of the inoculated seeder groups compared to day 7 PI. The aerosol-inoculated seeder chickens had a significantly higher average HI titer of 6.9 (log_2_ scale) on day 14 PI compared to the directly inoculated seeder chickens with an average HI titer of 4.1 (*p* < 0.05) (Figure 3). The results of HI titers suggest that inoculation via the aerosol route was a superior method when considering the induction of antibody-meditated immune responses against H9N2 AIV compared to the direct inoculation method.

Of the seeder chickens that were inoculated via the aerosol route, 93% had seroconverted by day 7 PI and 87.5% of these infected chickens were seroprotected by day 14 PI. In the directly inoculated seeder group, 81% of the seeder chickens had seroconverted by day 7 PI and 45% of the seeder chickens were seroprotected by day 14 PI. The seroprotection rate on day 14 DPI was significantly higher in the aerosol-inoculated seeder chickens compared to the directly inoculated seeder chickens (*p* < 0.05). 

In the recipient groups, HI titers in the recipient chickens exposed to the aerosol-inoculated seeder chickens were higher, with average titers of 3.5 and 6 on days 7 and 14 PE, respectively, when compared to the recipient chickens exposed to the directly inoculated seeder chickens, where the average titers ranged from 2.5 to 4.1 on days 7 and 14 PE, respectively. HI titers in the recipient chickens exposed the aerosol-inoculated seeder group was two-fold higher compared to the recipient chickens of the directly inoculated seeder group on day 7 PE, and the difference between the titers increased to 3.7-fold by day 14 PE (Figure 4). Furthermore, 87% of the recipient chickens seroconverted in both the aerosol and direct inoculation groups on day 7 PE. The recipient chickens that were infected via direct contact transmission with the aerosol-inoculated seeder chickens had a higher rate of seroprotection by day 14 PE when compared to the recipient chickens exposed to the directly inoculated seeder group. The rate of seroprotection was approximately 75% in the recipient chickens of the aerosol group whereas it was 37% in the recipient chickens of the direct inoculation group.

To determine the magnitude and nature of the association between virus load and HI titers in recipient chickens that were infected via direct contact transmission through aerosol or directly inoculated seeder chickens, a Spearman correlation test was used to estimate the correlation between the two parameters. The results suggest that there was a statistically significant and positive correlation between oral virus loads on days 3 and 5 PE and HI titers on day 14 PE in the recipient chickens exposed to the aerosol-inoculated seeder chickens (*p* < 0.05) (Figure 5 and Figure 6). In terms of cloacal shedding, a similar trend in correlation was observed between cloacal virus load only on day 5 PE and HI titers on day 14 PE in the recipient chickens of the aerosol group (Figure 6). 

## 4. Discussion

The H9N2 LPAIV subtype has become one of the most prevalent AIV subtypes in poultry over the last two decades. In addition to economic losses to the poultry industry, H9N2 AIV also poses a major public health risk for human health. The zoonotic potential to infect a wide range of species including humans has raised concerns [3,28]. However, there is a scarcity of information on how this virus is transmitted [21,29,30]. Therefore, the current study investigated the impact of different routes of inoculation on the transmission of H9N2 AIV in chickens by establishing a contact transmission model that closely simulates natural infection. 

Two different inoculation routes “i.e., aerosol and direct inoculation” were studied to establish infection and subsequent transmission of H9N2 AIV in chickens. The inoculated seeder groups were infected with H9N2 AIV and a group of healthy and uninfected recipient chickens were added to the same Horsfall unit 24 h post-inoculation of the seeder inoculated chickens. Both groups were kept together for 2 weeks post-introduction of the recipient chickens with the inoculated seeder chickens. As expected for a virus with low pathogenicity, the infected chickens (inoculated seeders and exposed recipients) remained asymptomatic, with no overt signs throughout the infection period. A defined virus dose for a sustainable transmission of LPAIVs remains unknown, due to variations in adaptability of LPAIVs to different species [21]. Previous studies have revealed that the replication rate of AIV differs among species, usually being higher in the species of origin. For instance, AIV isolated from turkeys replicates poorly in chickens [21,22,23]. Since the virus used in this study had a turkey origin, we decided to use a higher inoculation dose to ensure infection with high virus shedding and transmission to recipient chickens. 

The results from the current study reveal that chickens can be experimentally infected with H9N2 AIV by direct inoculation as well as the aerosol route. Maximum virus shedding from the infected chickens was observed on days 3 and 5 PI from the oral and cloacal routes, respectively, providing evidence for effective replication of H9N2 AIV within the infected chickens. This finding is in agreement with reports from previous studies, which state that peak replication of H9N2 AIV occurs between days 4 and 6 PI and declines by day 10 PI [31,32]. Furthermore, given the higher rate of infection and oral and cloacal shedding from the aerosol-inoculated seeder chickens compared to the directly inoculated seeder chickens, it is possible that the aerosol route of inoculation was more efficacious for experimental infection of chickens compared to the direct inoculation route. Previous studies by Swayne and Slemons (2008) found that poultry species are more susceptible to aerosolized H9N2 AIV infection when compared to intranasal, oral, or ocular routes [21]. Similar observations were also reported by Guan and colleagues (2013), who showed that chickens could be readily infected with aerosolized H9N2 AIV within short distances in a housing unit [16]. Together, these findings suggest that inoculation with H9N2 AIV via the aerosol route is more effective at establishing infection compared to the direct inoculation route when using a particular dose of virus inoculation. However, it is noteworthy that the susceptibility of chickens to H9N2 AIV varies with the route and dose of inoculation. For example, in a previous study by Yao and colleagues (2014), a relatively higher dose of H9N2 AIV was required for administration via the oral route compared to the dose required for establishing infection via aerosols [33].

The results from the present study highlight a relatively high virus load in the oral and cloacal swabs from the aerosol-inoculated seeder chickens. However, after comparison of virus titers in oral and cloacal swabs, it was apparent that shedding was higher via the cloacal route in all infected (inoculated seeder) chickens irrespective of the infection route and time point. This finding was in alignment with previous studies, which found an efficient replication of AIV in the respiratory tract when administered via the aerosol route compared to delivery of the virus in the form of large droplets [16]. Potential differences in replication of LPAIV in the respiratory and intestinal tracts can be attributed to the differential distribution of sialic acid receptors in these sites [34]. Additionally, the anatomical slit present in the upper crest of the palate, which separates the nasal cavity from the oral cavity, may allow a portion of the virus inoculum that is delivered intranasally (direct inoculation route) to enter the oral cavity and be swallowed into the gastrointestinal tract (GIT). Some studies have shown that a portion of the inoculum, when administered through the ocular route, can flow through the nasolacrimal duct into the nasal cavity and is eventually swallowed as well [35]. On the other hand, the minute size of the aerosolized virus particles generated by nebulizing the virus inoculum (<5 microns) facilitates the ability of the virus particles to travel into the lower respiratory tract, causing efficient replication and infection of the respiratory epithelial cells [36]. Studies suggest that aerosol particles with a size between 5 to 20 microns can remain suspended in the air for as long as 4–5 min and can gain entry into the nasal cavity, whereas virus particles in the droplet form are trapped in the external nasal cavity and are incapable of migrating deeper into the respiratory tract to infect respiratory epithelial cells [37]. 

Another plausible reason for higher cloacal shedding in the infected chickens can be attributed to infection in chickens via oral-fecal [38] or cloacal routes [39]. Due to the consistent excretion of the virus in fecal droppings, it is possible that AIV can gain entry into the GIT tract via oral-fecal or by cloacal routes. Hauck and colleagues (2017) have previously demonstrated that LPAIVs can persist in water in addition to fecal and bedding materials for more than 12–24 h [25]. A report by Meijer and colleagues (2004) also reported that the permeability of the cloacal membrane and the continuous contractile movements in the distal portion of the gastro-intestinal tract can act as a plausible route of AIV uptake and account for virus replication and cloacal shedding in the distal portion of the intestinal tract [39]. Some studies have also highlighted a ‘fecal-nasal’ route of transmission, suggesting that virus particles trapped in the exterior nasal cavity pass through the anatomical slit into the GIT through fecal-contaminated feed and infect the intestinal cells [18].

Previous studies have suggested the importance of virus-laden objects or surfaces that may mediate an ‘indirect’ transmission of AIV to a susceptible population [40]. In the present study, when the aerosolized H9N2 AIV was delivered to chickens, there is a possibility that some of the aerosols may have been deposited onto the chicken’s body, feathers, shank, and beak, and eventually may have been ingested by other chickens due to their pecking behavior. The ingestion of H9N2 AIV through indirect routes might also affect an overall virus shedding by the aerosol-inoculated seeder chickens and therefore this may be considered to be limitation of the present study. 

Based on the observed results of transmission in the recipient groups, the present study reveals that both aerosol and direct inoculation routes of infection were efficient in establishing a contact-based transmission model of H9N2 AIV from the inoculated seeder chickens to the exposed recipient chickens. Horizontal transmission of H9N2 AIV occurred more readily in the recipient chickens from the inoculated seeder chickens that were initially infected via the aerosol route compared to the direct inoculation route. It was previously reported that the aerosols may play a prominent role in AIV infection in chickens [38]. In the present study, the recipient chickens exposed to the directly inoculated seeder group were found to be positive for H9N2 AIV infection, although virus titer was lower at various sampling points compared to the recipient chickens exposed to the aerosol-inoculated seeder group. A higher rate of infection was observed in the recipient chickens of the aerosol group, demonstrated by a high virus load in oral and cloacal swabs and a greater number of chickens infected at each time point. The higher transmission of H9N2 AIV in the recipient chickens exposed to the aerosol-inoculated seeder chickens can be attributed to a higher percentage of chickens infected initially in the aerosol-inoculated seeder group. It can be assumed that the greater percentage of infected chickens might have led to a greater surface area of virus dissemination and increased chances of contact-based transmission of H9N2 AIV. 

Serum antibodies against AIV were analyzed as an additional evidence for the establishment of infection. Our findings suggest that the aerosol route of inoculation was a superior inoculation method when considering the induction of antibody-meditated immune responses against H9N2 AIV. The correlation between oral or cloacal shedding with the HI titers (day 14 PE) may partly explain this observation analyzed between virus load and antibody titers in the H9N2-AIV-infected recipient chickens exposed to the aerosol-inoculated seeder chickens. This demonstrates the higher efficacy of the aerosol route to establish infection and hence induce a higher magnitude of antibody response when compared to the direct inoculation route. The higher magnitude of antibody response in the chickens infected via the aerosol route can be attributed to the increased exposure of aerosolized virus to respiratory epithelial cells, causing higher replication and enhanced induction of immune responses [36]. 

Another plausible scenario to explain the elevated antibody levels in the aerosol-inoculated seeder chickens can be the nature of the innate responses induced by the aerosolized virus [41]. Previous studies suggest that the induction of the innate responses against respiratory viruses can direct the function of B cells present in different lymphoid aggregates in the respiratory tract [42]. It has been previously shown that type I interferons (IFN) orchestrate anti-viral responses in the respiratory tract and can influence the magnitude of antibody production in chickens [43,44,45]. Additionally, it is likely that the functionality of B cells varies with their location and interaction with innate immune cells, affecting antibody responses. 

In conclusion, the results from the present study reveal that the aerosol and direct inoculation routes can be used as effective methods for experimentally infecting chickens with H9N2 AIV. Moreover, it was shown that transmission of H9N2 AIV occurs more readily from chickens that were infected via the aerosol route as compared to the direct inoculation route. Future studies should focus on determining the relative contribution of different routes of transmission of LPAIV, the effect of various host factors (such as age or genetics), and factors related to the environment (e.g., temperature and humidity) and virus (such as dose and strain) that can affect the transmission dynamics of LPAIV.

## Figures and Tables

**Figure 1 vaccines-10-01040-f001:**
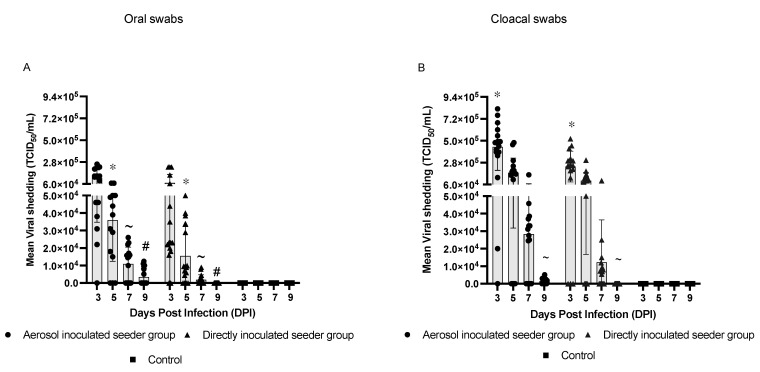
**Mean virus titers in inoculated seeder groups.** On day 14 of age, the inoculated seeder groups of the two experimental models were infected with H9N2 AIV either via the aerosol or direct inoculation route, or via PBS (control), respectively (n = 16/model). Virus load was assessed in oral (**A**) and cloacal swabs (**B**) based on TCID_50_ assay on days 3, 5, 7, and 9 PI. Each data point on the graph represents an individual chicken. Data were analyzed in using nonparametric Mann–Whitney unpaired Student’s t-test. Results were considered significant when *p* < 0.05. Symbols */~/# denote significant differences between the groups.

**Figure 2 vaccines-10-01040-f002:**
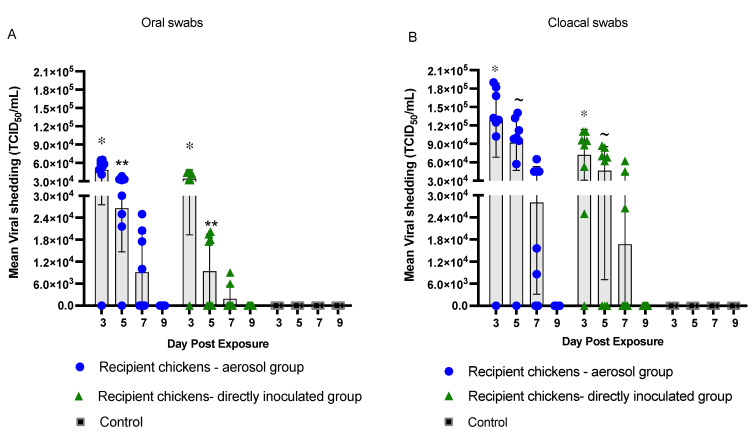
**Mean virus titers in the recipient groups.** Mean virus titers expressed as TCID50/mL in oral (**A**) and cloacal (**B**) swabs from the recipient chickens (n = 8/model) are shown on days 3, 5, 7, and 9 PE. Each data point on the graph represents an individual chicken. Twenty-four hours post-inoculation of the inoculated seeder groups, recipient (healthy and uninfected) chickens were added to the respective seeder groups. Data were analyzed using nonparametric Mann–Whitney unpaired Student’s t-test. The results were considered significant between the groups when */~ *p* < 0.05, or ** *p* value < 0.01.

**Figure 3 vaccines-10-01040-f003:**
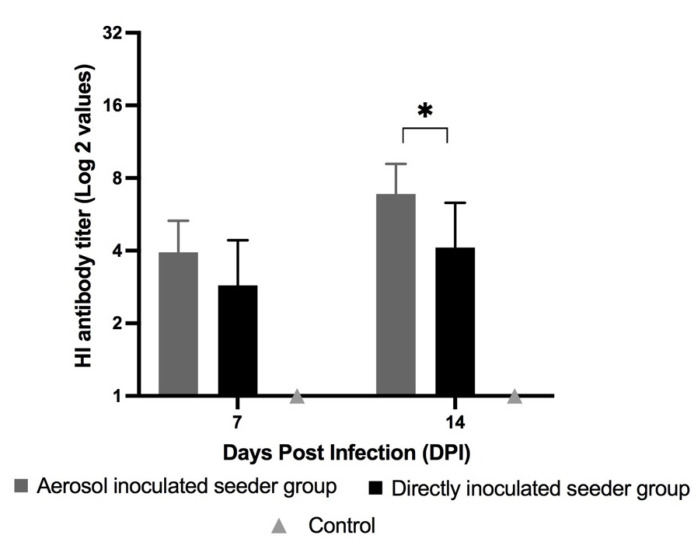
**Average serum HI antibody titers against H9N2 AIV in the inoculated seeder chickens.** On day fourteen of age, the chickens were inoculated either via aerosol or direct inoculation or via PBS (control), respectively (n = 16). Serum was collected on days 7 and 14 PI. The HI titers were first observed on day 7 PI. The control chickens remained negative on both time points. Data were analyzed using nonparametric Mann–Whitney unpaired Student’s t-test and were considered significant (*) when *p* < 0.05.

**Figure 4 vaccines-10-01040-f004:**
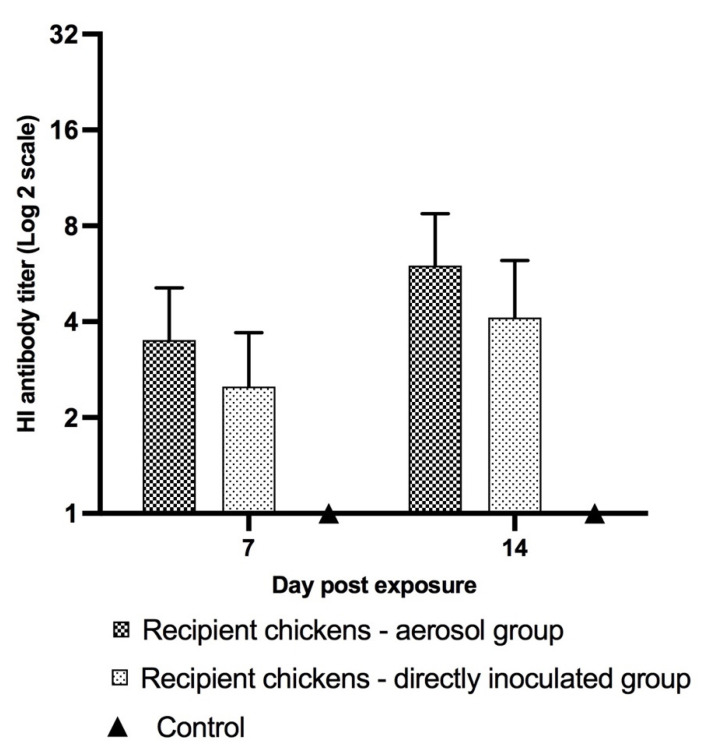
**Average HI antibody titers against H9N2 AIV in the recipient chickens.** Serum antibody titers in recipient chickens in both transmission groups are shown on days 7 and 14 PE. Twenty-four hours post-inoculation of the inoculated seeder groups, a group of recipient (healthy and uninfected) chickens were grouped with the respective seeder groups. Serum was collected on days 7 and 14 PE and analyzed for HI titers. Data were analyzed using nonparametric Mann–Whitney unpaired Student’s t test.

**Figure 5 vaccines-10-01040-f005:**
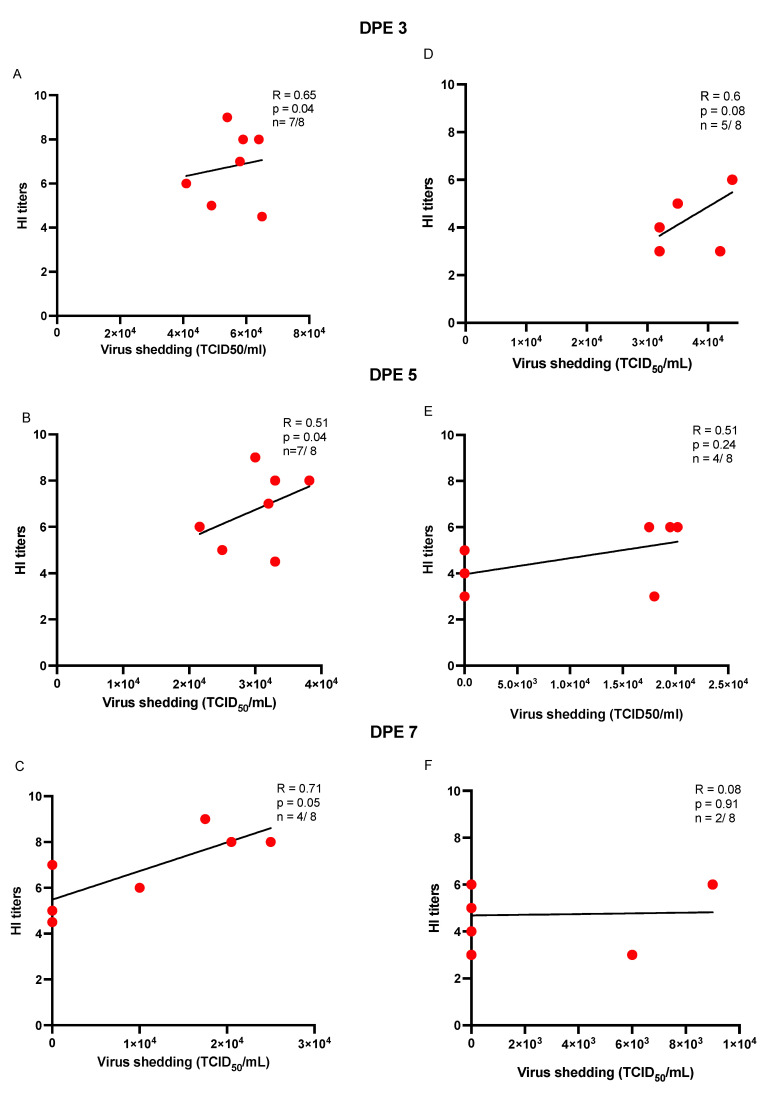
(**A**–**F**) Scatter plots illustrating the correlation of HI antibody titers and oral virus shedding in recipient chickens (expressed as TCID50/mL). Spearman’s correlation (R) was determined between oral virus shedding on days 3, 5, and 7 post-exposure (PE) with the HI titers (day 14 PE) in the H9N2-AIV-infected recipient chickens (n) of the aerosol (**A**–**C**) and direct inoculation (**D**–**F**) groups. The data points on the graph represent individual chickens that were deemed positive using TCID_50_ assay (out of the total recipient chickens) at the above-mentioned time points of swab collection.

**Figure 6 vaccines-10-01040-f006:**
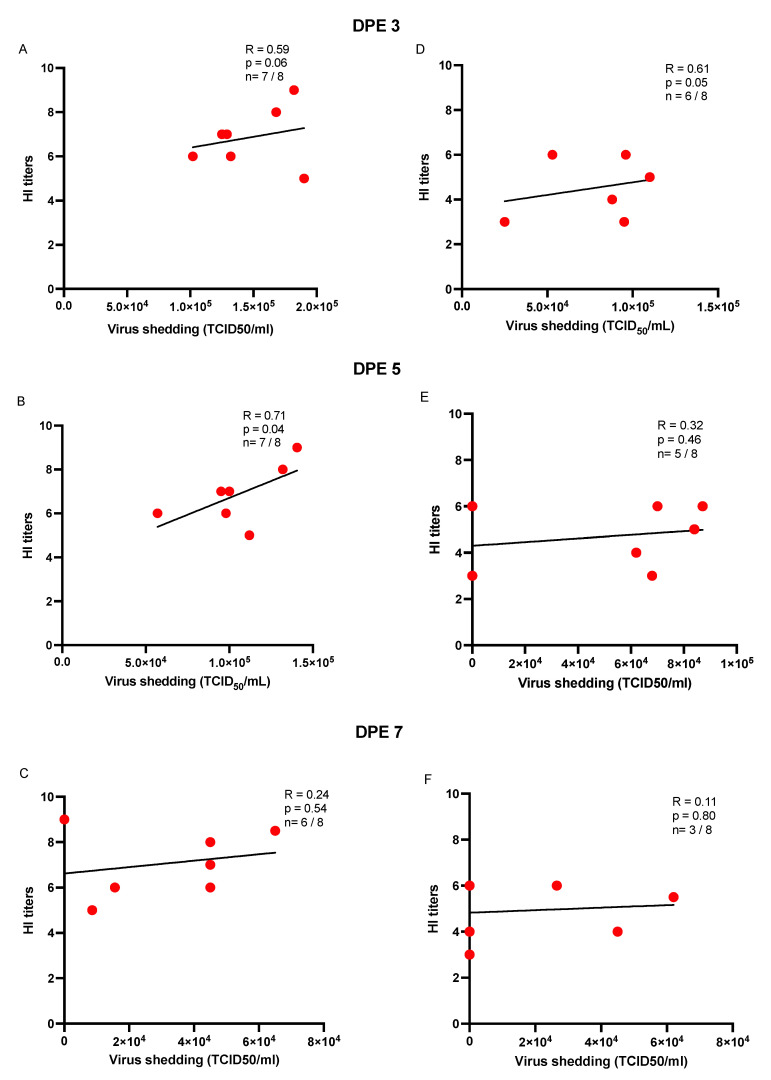
(**A**–**F**) Scatter plots illustrating the correlation of HI antibody titers and cloacal virus shedding in exposed recipient chickens (expressed as TCID50/mL). Spearman’s correlation (R) was determined between cloacal virus shedding on days 3, 5, and 7 post-exposure (PE) with the HI titers (day 14 PE) in the H9N2-AIV-infected recipient chickens (n) of the aerosol (**A**–**C**) and direct inoculation (**D**–**F**) groups. The data points on the graph represent individual chickens that were deemed positive using TCID_50_ assay (out of the total recipient chickens) at the above-mentioned time points of swab collection.

**Table 1 vaccines-10-01040-t001:** Effect of route of inoculation on H9N2 AIV infection in inoculated seeder chickens.

No. of Swabs Positive/No. of Swabs Tested				
Swabs (DPI)	Aerosol		Direct Inoculation	
**Oral**	**Birds positive**	**Mean ^1^**	**Birds positive**	**Mean ^1^**
**3**	15/16	1.20 × 10^5^	14/16	7.10 × 10^4^
**5**	13/16	3.60 × 10^4^	12/16	1.50 × 10^4^
**7**	10/16	1.10 × 10^4^	9/16	2.00 × 10^3^
**9**	6/16	3.50 × 10^2^	0/16	-
**Cloacal**	**Birds positive**	**Mean ^1^**	**Birds positive**	**Mean ^1^**
**3**	15/16	4.40 × 10^5^	11/16	2.40 × 10^5^
**5**	12/16	1.80 × 10^5^	10/16	9.80 × 10^4^
**7**	10/16	2.80 × 10^4^	5/16	1.20 × 10^4^
**9**	8/16	1.50 × 10^3^	0/16	-

^1^ Mean virus titers in the oral and cloacal swabs expressed in TCID_50_/mL

**Table 2 vaccines-10-01040-t002:** Rate of infection in recipient chickens of aerosol and direct inoculation groups.

No. of Swabs Positive/No. of Swabs Tested				
Swabs (PE)	Aerosol		Direct Contact	
**Oral**	**Birds positive**	**Mean ^1^**	**Birds positive**	**Mean ^1^**
**3**	7/8	4.80 × 10^4^	5/8	3.4 × 10^4^
**5**	7/8	2.60 × 10^4^	4/8	9.4 × 10^3^
**7**	4/8	9.10 × 10^3^	2/8	1.8 × 10^3^
**9**	0/8	-	0/8	-
**Cloacal**	**Birds positive**	**Mean ^1^**	**Birds positive**	**Mean ^1^**
**3**	7/8	1.30 × 10^5^	6/8	7.30 × 10^4^
**5**	7/8	9.20 × 10^4^	4/8	4.60 × 10^4^
**7**	6/8	2.80 × 10^4^	3/8	1.60 × 10^4^
**9**	0/8	-	0/8	-

^1^ Mean virus titers in the oral and cloacal swabs expressed in TCID_50_/mL

## Data Availability

Not applicable.

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
