# Peer review of "Transmission of H9N2 Low Pathogenicity Avian Influenza Virus (LPAIV) in a Challenge-Transmission Model"

_vaccines, 2022, doi:10.3390/vaccines10071040_

Round 1

Reviewer 1 Report

The researchers aim to address whether a direct contact model can be used to investigate the transmission models (i.e. direct inoculation versus aerosol) of LPAIV H9N2 virus.

The topic is relevant in the field. In spite of its LPAIV status, H9N2 AIV is the most widespread strain through the world with the ability to cause human infection and to donate virus segments to other AIV subtypes with severe downstream consequences.

Compared with other published material in the subject area, the transmission dynamics between seeder and recipient birds has not been investigated as thoroughly before.

The authors may want to consider whether molecular methods (real-time RT-PCR) can be used in conjunction with virus isolation to enhance their data sets.

This is an excellent, informative and well-written manuscript.

Author Response

Comment 1

The authors may want to consider whether molecular methods (real-time RT-PCR) can be used in conjunction with virus isolation to enhance their data sets.

Response 1

We thank the reviewer for the suggestion and will try to include the molecular methods to our future transmission experiments in conjunction with virus isolation. In the current experiment, we did not use the real-time RT PCR method as we only focussed on quantifying the replicating H9N2 AIV virus particles from the seeder and recipient chickens in the oral and cloacal swabs.

Reviewer 2 Report

H9N2 subtype low pathogenic influenza A viruses pose a significant public health concern having capability to infect various species. The authors mentioned that they conducted this study to establish direct contact model by infecting the chicken directly or through aerosols. There are studies referenced by the authors that were conducted to study different aspects of H9N2 IAV transmissibility in chicken and other animal models (ref. 11, 12, 23 and 24).  Ku K. B et al. (Virology, 450-451(2014)316–323, ref 24) compared the transmissibility of H9N2 and H7N9 IAV between chicken and chicken & ferrets, respectively inoculated by aerosol or via intranasal route.Therefore, the authors must clearly highlight the novelty of the present study in abstract and introduction part.  

1. The study clearly shows that LPAIV H9N2 were shed in higher titers through cloacal route than oral route regardless of the method of infection in both seeder and recipient chickens. Further, the authors showed in figure 5 that there was a significant correlation between HI titers and oral virus shedding in recipient chickens infected from aerosol-infected seeders on all days. However, this trend was not observed in cloacal swab samples (Figure 6). The authors need to discuss these results and possible justification in discussion part.

Minor points:

 1. Page 13, lines 372-379: the authors should correlate this discussion with the particle size generated by nebulizer used in present study.  

2. Page 2, Material and Methods 2.2: Kindly add information whether TPCK trypsin was used?

3. Page 3: The authors should add little information on how seeder and recipient chickens were differentiated.

4. Figure 1 and Figure 2: it would be better to add titles on figure such as A: Oral swabs, B: Cloacal swabs for easy understanding of the authors.  

5. Figure 5 & 6: The figures A-F does not have any titles. It is not clear that day 3, 5 and 6 data is related to which figure. It would be better to add titles to the figures.

Author Response

Reviewer 2

Comment 1

H9N2 subtype low pathogenic influenza A viruses pose a significant public health concern having capability to infect various species. The authors mentioned that they conducted this study to establish direct contact model by infecting the chicken directly or through aerosols. There are studies referenced by the authors that were conducted to study different aspects of H9N2 IAV transmissibility in chicken and other animal models (ref. 11, 12, 23 and 24).  Ku K. B et al. (Virology, 450-451(2014)316–323, ref 24) compared the transmissibility of H9N2 and H7N9 IAV between chicken and chicken & ferrets, respectively inoculated by aerosol or via intranasal route. Therefore, the authors must clearly highlight the novelty of the present study in abstract and introduction part.  

Response 1

- We appreciate your comment. The novel aspect of our current study was to explore the relative effect of two different routes of inoculation (aerosol and direct inoculation) to infect seeder chickens and compare the potential of the inoculated seeder chickens to transmit H9N2 AIV when co-housed with exposed recipient chickens in a direct contact transmission model. We evaluated how the routes of inoculation can affect the transmission of H9N2 AIV within chickens. Based on your suggestion, we have modified the abstract for clarity of context regarding the novelty of the study.

We have also made modifications in the summary lines of the introduction to bring more clarity to the novel aspects of the present study.  

Comment 2

  The study clearly shows that LPAIV H9N2 were shed in higher titers through cloacal route than oral route regardless of the method of infection in both seeder and recipient chickens. Further, the authors showed in figure 5 that there was a significant correlation between HI titers and oral virus shedding in recipient chickens infected from aerosol-infected seeders on all days. However, this trend was not observed in cloacal swab samples (Figure 6). The authors need to discuss these results and possible justification in discussion part.

Response: We thank the reviewer for the comment. Yes, we observed a significant correlation between oral shedding only on days 3 and 5 PE and HI titers in the recipient chickens infected from aerosol-infected seeders. In terms of cloacal shedding, similar response was observed on day 5 PE in the recipient chickens infected from aerosol-infected seeders. This finding has been mentioned in the results section on page 9-10; line 358-363. There were significant differences between oral shedding on day 3 and 5 PE and HI titers (day 14 PE). We saw a similar response in correlation between cloacal shedding only on day 5 PE and HI titers (day 14 PE). The justification for the higher infection and high antibody induction with respect to the above results is mentioned on page 13; lines 500-523. Despite of higher overall cloacal shedding, if we consider an average of coefficient of correlations (R) of oral and cloacal shedding respectively for the days 3, 5 and 7 PE, the average R value appears be lower between cloacal shedding and HI titers when compared to average R value between oral shedding and HI titers for the recipient chickens infected from aerosol or directly inoculated-infected seeders.

Minor comments

Comment 1. Page 13, lines 372-379: the authors should correlate this discussion with the particle size generated by nebulizer used in present study.  

Response 1: As suggested by the reviewer, a statement has been added signifying the correlation. Page 13, line 461-463

Comment 2. Page 2, Material and Methods 2.2: Kindly add information whether TPCK trypsin was used?

Response 2: We thank the reviewer for the suggestion. We have added the desired information in the Material and Methods section under the subheading “Virus isolation and processing of swab samples”. Page 4; 168-172

Comment 3. Page 3: The authors should add little information on how seeder and recipient chickens were differentiated.

Response 3: Based on the reviewers’ comments, we have added this information on Page 3 under experimental design Page 3, line 122-124

Comment 4. Figure 1 and Figure 2: it would be better to add titles on figure such as A: Oral swabs, B: Cloacal swabs for easy understanding of the authors.  

Response 4: We have modified the figures (Figure 1 and 2) by adding titles.

Comment 5. Figure 5 & 6: The figures A-F does not have any titles. It is not clear that day 3, 5 and 6 data is related to which figure. It would be better to add titles to the figures.

Response 5: As per reviewer’s suggestions we have modified the figures (Figure 5 and 6) by adding titles.

Reviewer 3 Report

I have some minor comments (see below):

Abstract: first sentence “Avian influenza viruses (AIV) can infect poultry and humans” is misleading, as it suggests humans to  be the only non-avian hosts. Please modify as follows: “Wild birds are the major reservoirs for Avian influenza viruses (AIV), which also infect poultry and occasionally, a number of mammalian species, including humans” (or equivalent, rephrased sentence)

Introduction: I suggest to add some more information and references on knowledge about H9N2. For instance:

General info at lines 40 to 51:

- high mutation rate and geographical spreading resulted in the evolution of five phylogenetically characterized H9N2 subgroups, also showing specific fingerprints (Heidari A et al., Sci Rep. 2018 Jan 31;8(1):1929. doi: 10.1038/s41598-018-20225-3. PMID: 29386534)

- mammalian adaptation markers have been reccently identified (Sun X et al. Viruses. 2020 May 14;12(5):541. doi: 10.3390/v12050541. PMID: 32423002

Airborne transmission of H9N2 at lines 54 to 60:

-  it has been reported also in mammals (Cáceres CJ et al. Viruses. 2021 Sep 24;13(10):1919. doi: 10.3390/v13101919. PMID: 34696349

In addition to the three references suggested above, please add further, updated info on H9N2, as this can be of help  to non specialist readers for defining relevance of the context (a journal such as Vaccines publishes papers on several different pathogens and systems, so part of readers might be really non-expert about AIV in general and  H9N2 in particular)

Materials and methods:

- virus inoculum (line 87): was the final dose in 250 ul somehow optimized? Was it suggested by previous papers? (that are not cited). Please explain.

- chicken population densiti (lines 99-100) it could be interesting to know if population density in the aerosol chamber (three chickens in 20000 cm3) is comparable to the one found in poultry farms

Results:

your data concern PI analyses; however, all experiments were performed with a fixed dose for infection. I am not asking for repeating analyses with different initial doses for infection, but could you please discuss this point, providing readers with rationale underlying the virus amount chosen as a dose for infection. Was it inferred from previous investigations? You may eventually discuss in the Discussion section.

Author Response

Reviewer 3

Comment 1.  Abstract: first sentence “Avian influenza viruses (AIV) can infect poultry and humans” is misleading, as it suggests humans to be the only non-avian hosts. Please modify as follows: “Wild birds are the major reservoirs for Avian influenza viruses (AIV), which also infect poultry and occasionally, a number of mammalian species, including humans” (or equivalent, rephrased sentence)

Response 1:  We have rephrased the sentence in the abstract for more clarity and added the recommended information according to the reviewer’s suggestions.

Comment 2. Introduction: I suggest adding some more information and references on knowledge about H9N2. For instance:

General info at lines 40 to 51:

- high mutation rate and geographical spreading resulted in the evolution of five phylogenetically characterized H9N2 subgroups, also showing specific fingerprints (Heidari A et al., Sci Rep. 2018 Jan 31;8(1):1929. doi: 10.1038/s41598-018-20225-3. PMID: 29386534)

- mammalian adaptation markers have been reccently identified (Sun X et al. Viruses. 2020 May 14;12(5):541. doi: 10.3390/v12050541. PMID: 32423002

Airborne transmission of H9N2 at lines 54 to 60

-  it has been reported also in mammals (Cáceres CJ et al. Viruses. 2021 Sep 24;13(10):1919. doi: 10.3390/v13101919. PMID: 34696349

In addition to the three references suggested above, please add further, updated info on H9N2, as this can be of help to non-specialist readers for defining relevance of the context (a journal such as Vaccines publishes papers on several different pathogens and systems, so part of readers might be really non-expert about AIV in general and H9N2 in particular)

Response 2: We appreciate the reviewer for this comment. We have attempted to address this by adding the following references.

  1. Heidari, A., Righetto, I. & Filippini, F. Electrostatic Variation of Haemagglutinin as a Hallmark of the Evolution of Avian Influenza Viruses. Sci Rep8, 1929 (2018). https://doi.org/10.1038/s41598-018-20225-3  (Page 1, line 42-44)
  2. Cáceres, C. J., Rajao, D. S., & Perez, D. R. (2021). Airborne Transmission of Avian Origin H9N2 Influenza A Viruses in Mammals. Viruses13(10), 1919. https://doi.org/10.3390/v13101919. 3 (Page 2, line 53-54)
  3. Righetto, I., & Filippini, F. (2020). Normal modes analysis and surface electrostatics of haemagglutinin proteins as fingerprints for high pathogenic type A influenza viruses. BMC Bioinformatics, 21(S10), 354. https://doi.org/10.1186/s12859-020-03563-w (Page 2, line 60-63)
  4. Sun, X., Belser, J. A., & Maines, T. R. (2020). Adaptation of H9N2 Influenza Viruses to Mammalian Hosts: A Review of Molecular Markers. Viruses12(5), 541. https://doi.org/10.3390/v12050541 (Page 2, line 60-63)

Comment 3 Materials and methods:

- virus inoculum (line 87): was the final dose in 250 ul somehow optimized? Was it suggested by previous papers? (that are not cited). Please explain.

Response 3-

We had conducted pilot studies in our lab using different doses of H9N2 AIV ranging from 2 x 106 , 2 x 107, 2 x 108, 4 x 108 TCID50units etc. The lower doses were effective at establishing infection in the seeder inoculated chickens, however we did not observe any shedding via oral and cloacal routes in the recipient groups at the lower doses. With respect to the volume used in the present study, we followed the previously used volumes mentioned in the literature for inoculation in chickens with slight modifications

(Barjesteh et al., 2015; França et al., 2012; Umar et al., 2016)

Comment 4 - chicken population densities (lines 99-100) it could be interesting to know if population density in the aerosol chamber (three chickens in 20000 cm3) is comparable to the one found in poultry farms

Response 4 -We agree with the reviewer’s comment. It will be interesting to look at the effect of population densities on the transmission of H9N2 AIV in chickens at the farm level and whether small scale models can mimic field conditions to help mitigate the outbreaks. We will definitely try to include this aspect to our future transmission experiments.

Comment 5 Results: your data concern PI analyses; however, all experiments were performed with a fixed dose for infection. I am not asking for repeating analyses with different initial doses for infection, but could you please discuss this point, providing readers with rationale underlying the virus amount chosen as a dose for infection. Was it inferred from previous investigations? You may eventually discuss in the Discussion section.

Response 5 – We appreciate the reviewer for this comment. As, suggested we have added this information to the materials and methods (page 2; line 100-117) and discussion section (page 12 ; line 409-413) . Prior to this experiment, we had conducted several experiments in our lab to establish a direct-contact transmission model, for example by titrating different infectious doses, placing seeder and recipient chickens in two adjacent Horsfall isolators connected through their air outlet and inlet, respectively. The results from our pilot studies revealed that the lower doses previously used for inoculation (Gonzales et al., 2011) were efficient at causing infection in seeder inoculated chickens however, the doses were ineffective in establishing transmission and subsequently infect the recipient chickens. We tried to establish transmission using different doses, ranging from 2 x 106, 2 x 107, 4 x 108, 2 x 108  and 8 x 108 TCID50 units. A defined virus dose for a sustainable transmission of LPAIVs remains unknown, due to variations in adaptability of LPAIVs to different species. Previous studies have revealed that replication rate of AIV differs among species, being usually higher in the species of origin. For instance, AIV isolated from turkeys replicates poorly in chickens. Since the virus used in this study had a turkey origin, we tested different doses (data not shown) and decided to use a higher inoculation dose to ensure infection with high virus shedding and transmission to recipient chickens.

Reviewer 4 Report

This appears to be a sound piece of work. The review is appropriate and the language of the paper is appropriate.

Minor comments

Line 18 The Aerosol inoculated chickens did not show shedding. The shedding was observed.

Line 340 Similarly the maximal vial shedding was observed. The chickens did not actively participate in demonstrating this phenomenon.

Line 411 Again they did not show.

Author Response

Reviewer 4

We thank the reviewer for his suggestions and comments.

Minor comments

Comment 1 Line 18 The Aerosol inoculated chickens did not show shedding. The shedding was observed.

Response 1- As suggested by the reviewer, we have rephrased the statement on page 1; line 19

Comment 2 Line 340 Similarly the maximal vial shedding was observed. The chickens did not actively participate in demonstrating this phenomenon.

Response 2- As suggested by the reviewer, we have rephrased the statement on page 9; line 358

Comment 3 Line 411 Again they did not show.

Response 3- We have rephrased the sentence for clarity of context Page 13; line 500.

Reviewer 5 Report

Transmission of H9N2 low pathogenicity avian influenza virus (LPAIV) in a challenge-transmission model

By S Raj et al (Corresponding author: S Sharif)

Submitted to Vaccines (Editorial No. vaccines – 1731764)

General Comments

This manuscript contains a comparison of transmission characteristics of H9N2 low pathogenicity avian influenza virus (LPAIV) from seeder to recipient chickens depending on the route of inoculation of seeder chicken. Those were inoculated either directly (by a combination of the ocular, nasal and intra-tracheal routes) or by aerosol. It is claimed that inoculation by aerosols was more effective in establishing infection in seeder chickens than direct inoculation, that aerosol-inoculated chickens showed higher amounts of virus shedding and that recipient chickens held together with aerosol-inoculated seeder chickens had higher oral and cloacal virus shedding than recipient chickens housed with the directly inoculated seeder chickens. In addition, the specific antibody responses (measured by HI test) were higher in seeder and recipient chickens of the aerosol infected compared to the directly infected seeder chicken group.

While the primary data appear to be obtained diligently and to be reliable, on the whole the data are not very novel [see refs. 11, 32]. Furthermore, some of the differences stated are not convincing, and their significance should be rechecked. Some of the data presentation should be reconsidered. For its content, the manuscript is much too long. For its content, the manuscript is much too long.

Specific Comments

Line

16        The modalities of inoculation by direct routes should be spelled out in detail.

34        A reference should be provided for the differentiation of HPAIV and LPAIV.

78        Since the virus used for the experiments was isolated from turkey, the minimal infectious dose for chicken could differ and should be determined. See lines 331-337 of Discussion.

115      Consider phrasing: … inhaled by the aerosol inoculated seeder chickens…

132      Replace ‘rpm’ by ‘g’ number.

143      Consider phrasing: … the reciprocal of the highest dilution showing complete inhibition…

173f     Table 1. The numbers compared are unlikely to differ by chi square test.

179f     Fig. 1. The significance of differences in mean virus load in oral and cloacal swabs is not convincing, due to the narrow calibration of the ordinate. It should be considered to present the data as geometric mean ± SE in a tabulated form and subject the data to t tests.

211f     Fig. 2. See comment line 179.

230      Table 2. See comment line 173.

251f     Fig. 3. The data of Fig. 4 (line 274) are more convincing than those of Fig. 3.

295f     Figs. 5 and 6. The data of the figures are partial and appear to be grossly over-interpreted.

367      A reference should be provided for the differential distribution of sialic acid receptors in respiratory and intestinal tissues.

385      … A report by Meijer and…

436f     Due to lack of relevant data in the manuscript, this statement is weak and should be considered for omission.

467f     References

            Refs. 7 and 20 are incomplete. 

Author Response

Reviewer 5

General comments

General comments

This manuscript contains a comparison of transmission characteristics of H9N2 low pathogenicity avian influenza virus (LPAIV) from seeder to recipient chickens depending on the route of inoculation of seeder chicken. Those were inoculated either directly (by a combination of the ocular, nasal and intra-tracheal routes) or by aerosol. It is claimed that inoculation by aerosols was more effective in establishing infection in seeder chickens than direct inoculation, that aerosol-inoculated chickens showed higher amounts of virus shedding and that recipient chickens held together with aerosol-inoculated seeder chickens had higher oral and cloacal virus shedding than recipient chickens housed with the directly inoculated seeder chickens. In addition, the specific antibody responses (measured by HI test) were higher in seeder and recipient chickens of the aerosol infected compared to the directly infected seeder chicken group.

While the primary data appear to be obtained diligently and to be reliable, on the whole the data are not very novel [see refs. 11, 32]. Furthermore, some of the differences stated are not convincing, and their significance should be rechecked. Some of the data presentation should be reconsidered. For its content, the manuscript is much too long.

Response - We thank the reviewer for his comments and suggestions. We have made some significant modifications in the entire manuscript to enhance its readability and hope that the revised manuscript is in a shape that is acceptable. In terms of novelty, we have incorporated the aspects in the abstract and summary of the introduction. We have reanalyzed our data and reconfirmed the significance levels of the data presented. We have added the means of the oral and cloacal shedding for both the inoculated seeder and recipient groups (Table 1 and 2). We also tried to concise the manuscript as much as we can.

With respect to Ref 32, the researchers have compared the relative transmission routes/exposures in poulet, roosters and broilers. Additionally, in reference 11 by Guan et al 2013, reseachers have focussed on comparing three inoculation routes to establish infection in seeder chickens and then employed the intranasal route to establish a direct and indirect contact model. Thus, based on the findings of previous studies we aimed to determine the potential of aerosol route (due to its smaller particle size) to establish infection in seeder chickens. We further compared the aerosol route and direct inoculation route (combination of ocular, nasal and tracheal) to infect the seeder chickens and subsequently transmit virus to recipient chickens in a direct contact transmission model.

Specific Comments

Comment 4 Line 16    The modalities of inoculation by direct routes should be spelled out in detail.

Response 4- We apologize as we could not include the entire information due to the word limitation in the abstract section (200 words), but we have provided full detailed information about the direct inoculation method under material and methods section page 3; line 127-129

Comment 5 Line 34 A reference should be provided for the differentiation of HPAIV and LPAIV.

  Response 5 - We thank the reviewer for his suggestion. As, suggested we have inserted the reference on page 1, line 35.

Alexander, D. J. (2007). An overview of the epidemiology of avian influenza. Vaccine, 25(30), 5637–5644. https://doi.org/10.1016/j.vaccine.2006.10.05

78        Comment 6 Since the virus used for the experiments was isolated from turkey, the minimal infectious dose for chicken could differ and should be determined. See lines 331-337 of Discussion.

Response 6- We thank the reviewer for his comment. A defined virus dose for a sustainable transmission of LPAIVs remains unknown, due to variations in adaptability of LPAIVs to different species. Previous studies have revealed that replication rate of AIV differs among species, being usually higher in the species of origin. For instance, AIV isolated from turkeys replicates poorly in chickens. Since the virus used in this study had a turkey origin, we tested different doses (data not mentioned) and based on our results we decided to use a higher inoculation dose to ensure infection with high virus shedding and transmission to recipient chickens.

We conducted several experiments to establish a transmission model, for example by titrating different infectious doses. The results from our pilot study revealed that the lower doses previously used for infection in the literature (Gonzales et al., 2011) were efficient in causing infection in seeder inoculated chickens however, the doses were ineffective in establishing transmission and subsequently infect the recipient chickens. Therefore, we tested transmission using different doses (ranging from 2 x 106 , 2 x 107, 2 x 108 , 4 x 108  and 8 x 108 TCID50 units).

Comment 7 line 115 Consider phrasing: … inhaled by the aerosol inoculated seeder chickens…

Response 7- The above statement has been rephrased according to the reviewer’s recommendations. Page 3, line 146

Comment 8 line 132   Replace ‘rpm’ by ‘g’ number.

Response 8- The centrifugation speed has been calculated and converted accordingly from rpm to g number. Page 4, line 172

            Comment 9 line 143 Consider phrasing: … the reciprocal of the highest dilution showing complete inhibition…

       Response 9- We thank the reviewer for the minor revisions. We have changed the sentence on page 4; line 163 accordingly.

Comment 10 173f   Table 1. The numbers compared are unlikely to differ by chi square test.

Response 10– We thank the reviewer for the suggestions. We rechecked and analysed the results of Pearson correlation test (chi square test) between HI and virus shedding. There were no changes in the results and discussion section.

Comment 11 179f     Fig. 1. The significance of differences in mean virus load in oral and cloacal swabs is not convincing, due to the narrow calibration of the ordinate. It should be considered to present the data as geometric mean ± SE in a tabulated form and subject the data to t tests.

Response 11 – We thank the reviewer for the suggestions. We have modified Table 1 and have added the geometric mean ± SE in the table. The data was subjected to t tests analysis and no changes in the results and discussion section were observed.

Comment 12 -line 211f   Fig. 2. See comment line 179.

Response 12 – We have modified Table 2 and added the geometric means of oral and cloacal shedding titers from the recipient chickens for clarity of context.

Comment 13 - line 230 Table 2. See comment line 173.

            Response 13– We have modified Table 2 and added the geometric means of oral and cloacal shedding titers from the recipient chickens for clarity of context.

Comment 14- 251f  Fig. 3. The data of Fig. 4 (line 274) are more convincing than those of Fig. 3.

Response 14 - We thank the reviewer for the important observation. We reanalysed and confirmed our results that Figure 3 representing the HI titers between directly inoculated and aerosol seeder inoculated chickens had significant difference on day 14 DPI. day 7 PI. The aerosol inoculated seeder chickens had significantly higher average HI titer of 6.9 (log2 scale) on day 14 PI compared to the directly inoculated seeder chickens with an average HI titer of 4.1 (P<0.05) (Fig. 3).

Comment 15  295f  Figs. 5 and 6. The data of the figures are partial and appear to be grossly over-interpreted.

Response 15- We apologize for the confusion. To bring clarity to the interpretation of the results of the scatter plots Figure 5 and 6, we have rephrased the sentences in the results section. Page 9; line 358-362

Comment 16 367   A reference should be provided for the differential distribution of sialic acid receptors in respiratory and intestinal tissues.

Response 16- As per reviewer’s suggestions, we have inserted a reference for the distribution of sialic acid receptors in respiratory and intestinal tissues. Page 13, line 455 , reference [34]

            Costa, T., Chaves, A.J., Valle, R. et al. Distribution patterns of influenza virus receptors and viral attachment patterns in the respiratory and intestinal tracts of seven avian species. Vet Res 43, 28 (2012). https://doi.org/10.1186/1297-9716-43-28

Comment 17 385   … A report by Meijer and…

Response 17- The correction has been incorporated on Page 13 line 474

Comment 18-line 436f Due to lack of relevant data in the manuscript, this statement is weak and should be considered for omission.

Response 18- As per reviewer’s suggestions, the above statement has been omitted from the discussion section of the manuscript. Page 14, line 542

Comment 19 467f References- Refs. 7 and 20 are incomplete. 

Response 19- The above references has been corrected and modified.

Round 2

Reviewer 5 Report

The authors have considered the reviewer's comments/suggestions carefully, and the manuscript has much improved.

In Tables 1 and 2 a legend should explain to which parameter the columns 'Mean' relate.

Author Response

Comments and Suggestions

The authors have considered the reviewer's comments/suggestions carefully, and the manuscript has much improved.

In Tables 1 and 2 a legend should explain to which parameter the columns 'Mean' relate."

Response- We thank the reviewer for their suggestions. As per reviewer’s suggestion we have added a line under Table 1 and Table 2 signifying the desired information about the parameter of the “Mean” mentioned in both the tables.